# Self-Management of Medication on a Cardiology Ward: Feasibility and Safety of the SelfMED Intervention

**DOI:** 10.3390/ijerph192416715

**Published:** 2022-12-13

**Authors:** Toke Vanwesemael, Laura Mortelmans, Koen Boussery, Sue Jordan, Tinne Dilles

**Affiliations:** 1Department of Nursing and Midwifery Science, Centre For Research and Innovation in Care (CRIC), Nurse and Pharmaceutical Care (NuPhaC), Faculty of Medicine and Health Sciences, University of Antwerp, 2610 Antwerp, Belgium; 2Department of Healthcare, Thomas More University College, 2500 Lier, Belgium; 3Pharmaceutical Care Unit, Faculty of Pharmaceutical Sciences, Ghent University, 9000 Ghent, Belgium; 4Department of Nursing, Swansea University, Singleton Park Swansea, Wales SA2 8PP, UK

**Keywords:** hospital, inpatients, medication, self-management, medication safety, medication errors

## Abstract

An intervention, SelfMED, was introduced to facilitate patient self-management of medication during hospitalization. This study aimed to evaluate the feasibility and safety of the SelfMED intervention. All patients in a cardiology ward in a Belgian regional hospital were assessed for suitability for inclusion, applying an evidence-based stepped assessment tool. Patients eligible for medication self-management and willing to participate were included in the study (i.e., consecutive sampling). Patients who self-managed their medication were closely monitored by nurses. The feasibility of medication self-management was evaluated by implementation and completion rates and the opinions of cardiologists. Safety was evaluated by medication administration errors and errors in patients’ registration of intake. Of 159 patients assessed for eligibility to self-manage medication in-hospital, 61 were included. A total of 367 medicines were self-managed. Pill counts showed 3 administration errors (0.8%), and on 6 occasions (1.7%) the patient’s registration of the intake was incorrect. SelfMED was deemed feasible within the hospital ward. In cardiologists’ opinions, SelfMED requires substantial time investment. In summary, SelfMED facilitated patient medication self-management in-hospital. As an essential step in the preparation for a full trial, this study showed it is feasible and safe to implement the intervention and identified some possibilities for refinement.

## 1. Introduction

The concept of patients self-administering their medication in hospitals is nothing new. It has been cited in the literature since 1959 and was defined as “bedside self-medication and self-administered medications” [1]. The London Audit Commission, the Society of Hospital Pharmacists of Australia (SHPA), the Royal Pharmaceutical Society (RPS), and the United Kingdom Nursing and Midwifery Council (NMC) have encouraged the implementation of self-administration of medication in hospitals [2,3,4,5]. However, different approaches to the concept are described. The SPHA describes the self-administration of medication as an approach to evaluate medication management of hospitalized patients to prevent medication-related problems after discharge. RPS focuses on medication self-administration as a transfer of responsibility depending on the patient’s ability to manage the tasks. Since the term ‘self-administration of medication’ seems to focus mainly on the act of administration of medication, the term self-management of medication (SMM) was introduced recently [6]. Patients are not only administering their medication but they are also supported in managing, storing, organizing, and reporting their self-managed medication in hospitals and receive education from healthcare professionals, including nurses, physicians, and hospital pharmacists [7].

In-hospital SMM has several implications for patients and healthcare providers [8,9,10]. Randomized controlled trials (RCTs) report that self-managing patients were more adherent than the control arms [11,12,13,14,15,16]. Several studies show that patients’ knowledge of their medicines significantly increases after implementing SMM [11,14,15,17,18]. Most patients and professionals hold a positive attitude towards SMM, and high satisfaction rates (90–100%) are reported [7,9]. Nevertheless, staff are concerned about increased workload, time spent educating patients, preparing medication, work stress, and a perception of increased medication errors, all due to SMM [7,9,19]. In contrast, healthcare providers indicate that the invested time is recouped, as less time is needed for dispensing and drug rounds [9]. A recent RCT confirmed the time-saving aspect: nurses spent less time on medication management and patient supervision in the self-administering arm than in the control arm [15]. 

The impact of SMM on medication safety is very important. Previous research, including healthcare professionals’ and patients’ perceptions, indicates that SMM could decrease medication errors, but healthcare providers emphasize possible medication abuse or misuse during self-administration [7,19]. A systematic review considered medication errors during hospitalization as an outcome of SMM: three studies showed a higher rate of medication errors, while nine studies reported fewer errors in the self-management arms than the control arms [9]. A recent RCT found reduced errors in the intervention arm and indicated that SMM is safe [20]. However, solid evidence of the effect on medication safety remains scarce. 

Although, in-hospital SMM seems to be a promising strategy to sustain the continuity of medication management, increase medication adherence, and reduce errors, only 22% of patients in Belgium self-manage their medication during hospitalization [6]. Yet, up to 40% of patients would be able to manage their medication by themselves [6]. When SMM is allowed, it seldom occurs in a structured, evidence-based way. A minority (18%) of hospital wards have an SMM protocol, and only 7% have an assessment available to determine patients’ abilities to self-manage [6]. Furthermore, the content of available guidelines is diverse, and SMM instruments are not validated [3,21,22,23]. Reports on the implementation of SMM in acute hospitals in the United Kingdom indicate that uptake is variable [3,24]. 

Therefore, the SelfMED intervention was developed and validated to facilitate SMM by patients in-hospital. This intervention includes an assessment to evaluate whether self-management is advisable (SelfMED assessment), a monitoring tool for a structured follow-up of patients self-managing their medication (SelfMED monitoring), and a guide for nurses to intervene when self-management problems occur (SelfMED support) [25]. A protocol (SelfMED procedure) clarifying the overall SMM process, interventional components, and respective responsibilities of healthcare providers and patients, offers implementation guidance.

During this study, the SelfMED intervention was implemented in a cardiology ward in a regional hospital in Belgium. The aim of the study was firstly to report the feasibility of the SelfMED intervention and secondly to evaluate the safety of the intervention. Feasibility was assessed by the extent to which the SelfMED intervention was applied in practice (i.e., level of implementation) and a cross-sectional survey of cardiologists. Safety was assessed by medication administration errors and errors in patients’ registration of medication intake. This study is essential before the intervention can be subjected to a randomized controlled clinical trial. 

## 2. Materials and Methods

### 2.1. Design

A feasibility and safety evaluation of an SMM intervention was conducted in a regional hospital (Belgium). Feasibility was evaluated based on the completeness of implementation and a cross-sectional survey of cardiologists. Safety was assessed by medication administration errors and errors in patients’ registration of medication intake. 

### 2.2. Participants

The current SelfMED procedure focuses on the average hospitalized patient in a cardiology ward. Therefore, in a non-profit regional hospital (581 beds), all patients hospitalized in a cardiology ward (32 beds)—specializing in heart failure, cardio rehabilitation, and post-interventional care—were screened for eligibility in the study by the head nurse or treating nurse and treating physician using the SelfMED assessment (see further). All patients with a positive assessment for in-hospital medication self-management were invited to participate in the study (i.e., consecutive sampling). Patients aged < 18 years or unable to sign informed consent were excluded. Figure 1 shows the participant flowchart. 

### 2.3. SelfMED Intervention

The SelfMED flowchart is shown in Figure 2. First, patients were assessed to decide whether they were eligible for self-managing medicines in-hospital (SelfMED assessment). The first step of the assessment was performed by the nurse. A ten-statement assessment allows the nurse to assess the patient’s eligibility for SMM. If the nurse deems the patient eligible, the patient completes a written self-assessment. Based on these data, the nurse advises the treating physician or responsible medical officer. The treating physician makes the final decision regarding self-management and decides which medicines can be self-managed. Oral medications and inhalation therapies are considered appropriate for medication self-management. The physician can decide not to allow medication self-management for specific acute or high-risk medications (e.g., opioids, benzodiazepines). Medicines that patients typically do not administer themselves in the outpatient setting (e.g., intravenous medication) are excluded from medication self-management and administered by nurses during the standard medication rounds. 

If a patient is allowed to self-manage, several precautions need to be taken: the nurse provides the patient with a medication scheme, the hospital pharmacy delivers medication, the nurse stores medication in the patient’s room, the nurse or physician documents the self-managed medication in the patient’s medical file. Also, the nurse instructs the patients about the medication schedule to ensure that they are aware of the type of medication, the time of administration, and the dosage of self-managed medication. When the self-managed medication is taken, patients are instructed to tick these off on the medication schedule at the time and date of administration (SelfMED monitoring tool). This allows nurses to follow up on patients’ self-management, detect problems, and, if necessary, provide tailored interventions. The development and validation of the SelfMED intervention were described in detail in a previous article [25]. 

### 2.4. Study Outcomes

First, the feasibility of the SelfMED intervention in the cardiology ward was evaluated based on the extent of implementation (i.e., the proportion of eligible patients using the SelfMED intervention in practice) and a cross-sectional survey for cardiologists. This self-developed survey covered: content (5 items), user-friendliness (1 item) and time investment (1 item) in closed questions (10-point scale; 1 = absolutely not satisfied to 10 = absolutely satisfied). Perceived consequences of the self-management of medication (3 items) were reported using a 10-point scale (1 = strongly disagree—10 = strongly agree). Free text comments could be made at the end of the questionnaire.

Second, safety was evaluated by medication administration errors and errors in the patients’ registration of medication intake. The National Coordinating Council for Medication Error Reporting and Prevention (NCC MERP) Taxonomy of Medication Errors was used to categorize medication administration errors [26]. Medication administration errors were calculated by a daily pill count by the head nurse or lead nurse for the ward. Every day, nurses manually reconciled the number of remaining pills. The nurses looked for discrepancies between the patients’ logs in the registration form, medication stored in their rooms, and the anticipated medication scheme every day. Any problem was discussed with the patients to identify the reason for non-adherence. Ticking off the medication scheme incorrectly was considered a registration error. This was evaluated by comparing the patients’ logs with the anticipated medication scheme and performing a pill count, e.g., if a patient had taken his/her medicine correctly but had not correctly ticked off the medicines scheme, this was identified by a correct pill count and an error on the medication scheme. The discussion with the patient could confirm this medication registration error. This mechanism of control does not prevent the intended non-adherence of patients. 

### 2.5. Data Collection

Data were collected from February to March 2016. The stepped SelfMED assessment forms provided data on the nurse assessment, the patient self-assessment (extended with a questionnaire on demographics and medication intake at home), and the physician’s decision on allowing or declining medication self-management in-hospital. The SelfMED monitoring tool collected data from the patient’s log of the medication scheme and the nurse’s pill count. There was close involvement of the healthcare providers (e.g., head nurses and cardiologists) in-hospital to implement the entire procedure and carry out all registrations. Data collection instruments used (questionnaires/assessments) are available as Appendix A. 

### 2.6. Data Analysis

The Statistical Package for Social Sciences (SPSS Inc., Chicago, IL, USA) was used to analyze the data. Z-scores were used to test the normality of data [27] and showed non-normality of age distribution. Descriptive statistics were used to describe patient and medication management characteristics, the feasibility of the SelfMED intervention, and the safety of the intervention in terms of number of medication administration and registration errors. Categorical data were described using frequency distributions. Continuous data were described using a mean value and standard deviation if normally distributed or using a median and full range (min–max) if not normally distributed. 

Inferential statistics were used to determine differences in patient and medication management characteristics between the groups ‘no self-management allowed’ and ‘self-management allowed’. Pearson Chi-squared test was used for nominal variables and the Mann–Whitney U test for continuous variables. A *p*-value of 0.05 was considered statistically significant. 

Self-managed medicines were coded using the Anatomical-Therapeutic-Chemical (ATC) classification for analysis. The ATC classification is an internationally accepted classification system for medicines maintained by the World Health Organization (WHO). The ATC-Classification substances are divided into different groups according to the organ or organ system they affect and their chemical, pharmacological, and therapeutic properties [28]. 

## 3. Results

### 3.1. SelfMED Intervention—Feasibility

#### 3.1.1. Level of Implementation: Assessment

In total, 159 eligible patients were admitted to the cardiology ward (See Figure 1). During the first step of the intervention, these patients were assessed by nurses for eligibility, resulting in 85 patients considered eligible for self-management. Of these patients, 11 did not consent to participate. The remaining 74 patients completed the patient assessment. This second assessment showed that two patients were not willing to self-manage. Ultimately, 72 patients were eligible for and willing to self-management of medication. For one patient, the self-management of medication was not further considered due to early discharge. Based on the nurse assessment and the self-assessment, nurses advised physicians to allow 70 of the resulting 71 patients to perform self-management of medication. The treating physician could reflect on the nurse assessment, self-assessment, and the nurse’s advice during the decision-making process. The treating physicians allowed 61 patients to self-manage their medication, of whom 58 self-managed all their medication and three only some medicines. 

As presented in Table 1, the most prevalent reasons for patients not to pass the nurse assessment were: not being capable of handling changes in their medication regimen (n = 48, 49.0%), being mentally not able (n = 47, 48.0%), or not administering their medication at home (n = 44, 46.8%). Similarly, physicians reported the reasons for not allowing patients to self-manage during the final step of the assessment: a presumption of therapy non-adherence (n = 2), the physician does not trust the patient (n = 2), too many changes in the medication schedule (n = 2), and the patient did not take any medication at home (n = 1).

As described in Table 2, the median age of all screened patients was 75 years. Self-managing patients were, on average, 70 years old, 5 years younger compared to those who were not allowed to self-manage (*p* = 0.001, Mann–Whitney U). Of all patients screened, 47.5% were female and 52.5% were male. The majority of patients allowed to self-manage completed secondary school as their highest level of education.

Patients in the self-management intervention group used, on average, 4.5 different long-term medications at home and almost all (96.7%) completely self-managed these at home. One out of four of these patients had previously self-managed medication during hospital admission.

#### 3.1.2. Level of Implementation: Self-Management of Medication

A total of 367 medicines were self-managed in-hospital by 61 patients, with a median of 4 different self-managed medicines per patient over the entire duration of their hospitalization. The majority of these were medicines for the cardiovascular system (21.2%) and blood and blood-forming organs (8.9%). The median duration of self-management during hospitalization was 3 days (range 2–9 days) (See Table 3).

#### 3.1.3. Cross Sectional Evaluation by Cardiologists

As shown in Table 4, all cardiologists in this study completed a questionnaire on the feasibility of the SelfMED intervention. Overall, they scored a median of 6.0 out of 10 on their satisfaction with the SelfMED intervention and user-friendliness. A median score of 6.5 was provided on the content of the items of the nurse and patient assessment. Cardiologists agreed that self-management contributed to a better understanding of the patient’s competences to self-manage medication after hospital discharge (7.0/10 rating). 

The time investment to facilitate self-management scored lowest (4.0/10 rating). It was stated that teaching patients about self-management of medication takes time, and changes in the drug regimen can be rather complicated for patients. Therefore, cardiologists suggested the SelfMED intervention would be more relevant within wards with a longer hospital stay and relatively few changes in patients’ medication regimens. Because of the short hospital stay, cardiologists stated it was rather difficult to rate patients’ self-management skills, making it a challenge to provide feedback and follow-up on self-management. This was consistent with the rating of 6.5/10 on SelfMED, contributing to better self-management of patients prescribed medical therapy.

### 3.2. SelfMED Intervention—Safety

#### 3.2.1. Medication Administration Errors

In three cases, a medication administration error was reported. According to the NCC MERP taxonomy, the administration errors detected were dose omission (n = 2) and wrong duration (n = 1). One medication administration error concerned the medicine gabapentin (active substance: gabapentin), which was ticked off on the medicines scheme but not taken, as identified by pill count. The second administration error concerned a pill that was found on the floor by the nurse; the patient had ticked off all medication administrations. The third error concerned one patient who was informed of the decision to stop amiodarone treatment (active substance: amiodarone hydrochloride). The patient was administered this medicine one more time, as it was still available in the patient’s room. This medication administration error was identified based on the medication scheme and a pill count. Subsequently, amiodarone was removed from the patient’s room.

#### 3.2.2. Medication Registration Errors

Most errors were registration errors. They were the result of patients failing to tick off the medication (n = 4) or ticking incorrectly (n = 1), although the patient had taken the correct medicine at the right time. To prevent any recurrences, the patients were informed by a nurse about the use of the medication scheme. One patient ticked off the medication but did not take medicines according to the pill count.

Registration and administration errors totaled 9 in 367 self-managed medicines (2.5%). Nine different patients—14.8% of the self-managing population—made these errors. 

## 4. Discussion

### 4.1. Main Findings

During this evaluation, we assessed whether the SelfMED intervention is feasible and safe before using it in a large clinical trial. This study provided information on the functioning and application of the SelfMED assessment, the monitoring tool, the feasibility of the intervention within the ward, the effect of self-management on patient medication errors and a reflection on the entire process. The results of this evaluation study will be considered when preparing and installing a randomized controlled clinical trial with proper pre- and post-intervention measurements. Some adjustments will be made to the toolbox, such as the extension of the support tool to facilitate the implementation in clinical practice.

### 4.2. SelfMED Assessment

The SelfMED assessment—as a part of the SelfMED intervention—was implemented for the first time. Eleven patients did not consent to participate in the study, and two indicated in the self-assessment they were unwilling to self-manage their medicines. During the final step of the SelfMED assessment, physicians allowed 61 patients to self-manage (at least some of) their medication. The reasons for not allowing patients to self-manage are: the patient is incapable of handling changes in the regimen; the patient is mentally unable to self-manage (i.e., patients with cognitive impairment); the physician does not trust the patient; or the patient does not adhere to the therapy. Non-adherence should not be a reason for prohibiting SMM because evidence from RCTs shows a positive impact of medication self-management on medication adherence [14,15].

When patients return home, they are expected to be responsible for self-managing medication. As stated by the Society of Hospital Pharmacists of Australia, self-management of medicines in-hospital could identify and address problems as a part of the discharge planning process. The assessment and success rate of self-management can determine the needs for support after hospital discharge [2]. Therefore, the reasons for allowing or declining the self-management of medication should be comprehensively evaluated. Research on patients discharged from different wards (internal medicine, pulmonary medicine, neurology, and cardiology) indicated medication-related problems occurred in 18 out of 104 patients (15%) [29]. The problems were very diverse, i.e., patients did not know indications, how to administer medication, had concerns about combinations of medicines, did not receive patient information leaflets, or lacked prescriptions for their medicines therapy. This stresses the importance of guiding patients and supporting them to self-manage medication while in-hospital.

### 4.3. SelfMED Intervention—Feasibility

The evaluation of the feasibility of the SelfMED intervention indicated that the cardiologists were satisfied with the number of questions within the SelfMED assessment. Also, they agreed the intervention contributed to a better understanding of the patient’s competencies in self-management of medication after hospital discharge. This knowledge of the patient’s competencies is an advantage, given that patients experience problems such as a lack of knowledge, resources and self-efficacy regarding self-management after discharge and transitioning to their home environment [30].

Overall, the cardiologists suggested facilitating self-management in patients who remained hospitalized for longer periods, as teaching patients about their medicines takes time. These findings were in line with those reported in a systematic review by Richardson et al. (2014). Although teaching patients takes time, nurses will spend less time on medication preparation and administration. Also, patients successfully self-managing their medication will require a shorter time investment [9,15]. In our study, ‘time’ was based on the perception of the cardiologists. Determining the exact time investment for facilitating self-management with the SelfMED intervention requires additional research from the viewpoint of all stakeholders (i.e., physicians, nurses, patients, and hospital pharmacists). 

Previous research indicated that it is not only physicians who play an important role in the self-management of medication; nurses, patients and hospital pharmacists are also involved in the SelfMED intervention [6,25]. Therefore, further research should include these stakeholders when evaluating the feasibility and all components of the intervention (i.e., the SelfMED assessment, the SelfMED monitoring tool and the medication scheme) in daily practice.

This study was completely paper-based, as the medication management system on the ward did not allow the SelfMED intervention to be included within their medical software. SWOT analyses on self-management of medication in-hospital recommended using an electronic medical file when implementing medication self-management so that all healthcare providers have a constantly updated overview of self-managed and nurse-administered medication [7]. Therefore, we suggest that the SelfMED intervention is included in the electronic medical software of hospitals. This would simplify the assessment of patients, resulting in immediately available information for all healthcare providers. Moreover, patients could be provided with an electronic device or application to tick off their medicines (instead of the paper-based medication scheme). This information would be directly linked to the medical software package of the hospital.

### 4.4. SelfMED Intervention—Safety

The self-management of medication facilitated with the use of the SelfMED intervention resulted in low medication administration errors (0.8% of 367 self-managed medicines). A systematic review of descriptive studies on the self-management of medication reported error rates of between 2.5% and 7.5% in self-administering patients. However, comparing our results with error rates in the systematic review was difficult because of the lack of a precise definition of medication errors or a description of how they were observed [9].

During our study, medication administration errors were evaluated using the patient’s log and a pill count. Future research should use additional tools and supplementary observation of self-managing patients to evaluate medication administration errors (i.e., wrong time, wrong dose, wrong administration technique, or wrong medicine) more precisely. In order to compare care as usual and medication self-management it would require a control or comparator group receiving treatment as usual (nurse-administered medication) and an intervention group (medication self-management). These study results can provide more insights, i.e., on the effect of the SelfMED intervention on the safety of patients in relation to medication errors. Follow-up after hospital discharge should be included in future work.

Medication registration errors provided insight into the complexity of the SelfMED monitoring tool. Completing the registration form seemed relatively easy for patients, given the low error rate. However, to avoid registration errors in future studies, patients should be adequately educated on using the registration form as part of the monitoring system. 

### 4.5. Limitations of the Study

Firstly, possible reasons for unwillingness to participate in the study or to self-manage medication were not documented. Secondly, the SelfMED intervention was exclusively tested on one cardiology ward of a regional hospital, limiting the generalizability of the study results. Thirdly, the survey on the feasibility of the SelfMED intervention comprised only a small number of questions and was not validated beforehand. Fourthly, although all healthcare providers (nurses, physicians, and pharmacists) were invited to assess feasibility, only a limited number of cardiologists filled out the survey. Fifthly, the evaluation of time investment was based on the cardiologists’ perception and not on the exact timing of the different actions of medication management. The study was conducted pre-COVID, and healthcare professionals’ workloads have increased since. This may impact the feasibility of SelfMED.

### 4.6. Practice Implications

Information on the functioning and application of the SelfMED assessment, the monitoring tool, the effect of self-management on patient medication errors, the feasibility of the intervention within the ward, and a reflection on the entire process is valuable. It should be taken into account when preparing and installing a complex controlled clinical trial. The potential of the SelfMED intervention should be explored in a long-term observational study to observe SMM post-discharge, in a restricted population, before proceeding to a multicentered controlled trial.

## 5. Conclusions

The SelfMED assessment identified eligible patients willing to self-manage their medicines in-hospital. The SelfMED intervention resulted in a low number of medication administration and registration errors. Comments regarding feasibility and time investment to facilitate self-management should be taken into account during future investigations. A multicentered randomized controlled trial is needed to provide solid evidence on the effectiveness of the SelfMED intervention, making it possible to make evidence-based decisions regarding the implementation of the intervention in daily practice.

## Figures and Tables

**Figure 1 ijerph-19-16715-f001:**
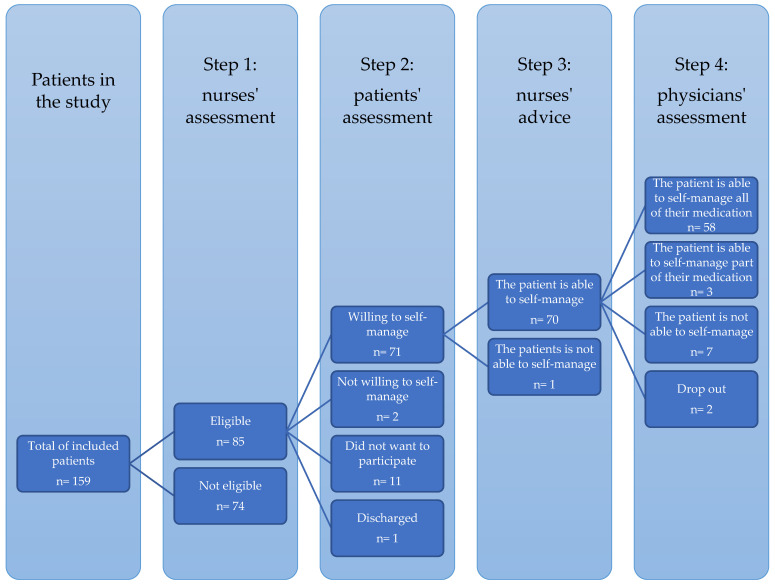
Flowchart of the participants in the study.

**Figure 2 ijerph-19-16715-f002:**
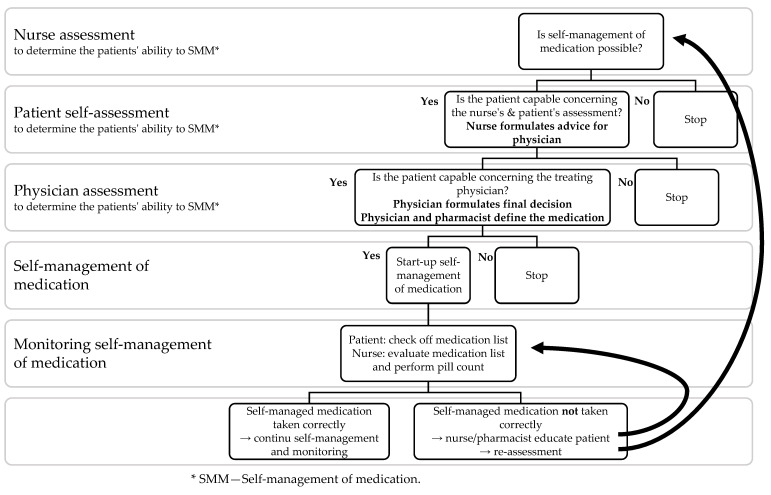
The SelfMED flowchart.

**Table 1 ijerph-19-16715-t001:** Results of the nurse assessment.

Questions in the Assessment	Positive Assessment ^a^ n = 61 % (n)			Negative Assessment ^b^ n = 98 % (n)		
	Agree	Dis- Agree	Un- Known	Agree	Dis- Agree	Un- Known
1. The patient can prepare his/her own medication at home.	98.3 (59)	1.7 (1)	0.0	51.1 (48)	46.8 (44)	2.1 (2)
2. The patient administers his/her own medication at home.	100 (61)	0.0	0.0	78.7 (74)	19.1 (18)	2.1 (2)
3. After discharge from hospital the patient is capable of preparing and administering his/her own medication.	98.3 (60)	1.7 (1)	0.0	54.3 (51)	44.7 (42)	1.1 (1)
4. The patient is physically able to administer his/her own medication.	98.4 (60)	1.6 (1)	0.0	67.3 (66)	32.7 (32)	0.0
5. The patient is mentally able of controlling his/her own medication.	100.0 (61)	0.0	0.0	52.0 (51)	48.0 (47)	0.0
6. To the best of my knowledge, the patient has no history of substance misuse.	88.5 (54)	9.8 (6)	1.6 (1)	43.3 (42)	7.2 (7)	49.5 (48)
7. The expectation is that the patient can handle the prescribed treatment regimen.	96.7 (59)	3.3 (2)	0.0	48.0 (47)	49.0 (48)	3.1 (3)
8. The patient will not be exposed to any other clinical interventions (i.e., surgery) where a new medication regimen will need to be implemented.	96.7 (59)	0.0	3.3 (2)	86.6 (84)	3.1 (3)	10.3 (10)
9. The expectation is that the hospitalization will be long enough to allow the patient to commence self-administering his/her own medication.	98.4 (60)	1.6 (1)	0	76.3 (74)	18.6 (18)	5.2 (5)
10. The patient speaks sufficient Dutch to understand the treatment (verbal, written).	100.0 (61)	0	0	91.8 (89)	4.1 (4)	4.1 (4)

^a^ Positive assessment: Based on the overall results of the nurse assessment, the nurse defines the patient eligible for self-managing medication. ^b^ Negative assessment: Based on the overall results of the nurse assessment, the nurse defines the patient not eligible for self-managing medication.

**Table 2 ijerph-19-16715-t002:** Patient demographic data and medication management characteristics.

	**All Screened Patients** **(n = 158)**	**Self-Management Not Allowed** **(n = 97)**	**Self-Management Allowed** **(n = 61)**	**Test Statistic**	***p*-Value**
Age (years)				U = 2065	0.001
mean ± SD	72.8 ± 13.6	75.0 ± 14.1	69.5 ± 12.0		
median (min–max)	75.0 (23–95)	79.0 (23–95)	71.0 (45–89)		
Gender (%)				X^²^ = 0.936	0.333
Female	47.5	50.5	42.6		
Male	52.5	49.5	57.4		
Level of education (%)				X² = 3.885	0.422
None	12.3	15.4	11.7		
Primary school	17.8	15.4	18.3		
Secondary school	52.1	69.2	48.3		
Bachelor level degree	15.1	0.0	18.3		
Master level degree	2.7	0.0	3.4		
Number of long-term medicines taken at home ^a^				-	-
mean ± SD			4.5 ± 3.2		
Medication management at home (%) ^a^				-	-
Self-management			96.7		
Aids used for preparation			3.3		
Aid for preparation and self-administration			0.0		
Self-management of medication during previous admission ^a^				-	-
Yes (%)			24.6		

^a^ Data not available for the group of patients not allowed to self-manage medication during hospitalization.

**Table 3 ijerph-19-16715-t003:** Assessment of self-managed medication.

Characteristics	Mean ± SD	Median (Min–Max)	%
Duration of patient self-management of medication (days) (n = 61)	3.7 (1.4)	3.0 (2–9)	
Number of self-managed medicines per patient during hospital stay (n= 367 ^a^)	5.0 (2.9)	4.0 (1–11)	
Main ATC group of self-managed medicines (n = 367 ^a^) ^b^			
A: Alimentary tract and metabolism			6.0
B: Blood and blood forming organs			8.9
C: Cardiovascular system			21.2
D: Dermatologicals			0.0
G: Genitourinary system and reproductive hormones			0.9
H: Systemic hormonal preparations, excluding reproductive hormones and insulins			1.0
J: Anti-infectives for systemic use			0.6
L: Antineoplastic and immunomodulating agents			0.1
M: Musculoskeletal system			1.0
N: Nervous system			3.3
P: Antiparasitic products, insecticides, and repellents			0.0
R: Respiratory system			0.9
S: Sensory organs			0.6

^a^ Total number of self-managed medicines. ^b^ Every self-managed medicine was categorized by its main anatomical group with the use of the Anatomical Therapeutic Chemical Classification System [28].

**Table 4 ijerph-19-16715-t004:** Feasibility of the SelfMED procedure as reported by participating cardiologists (n = 4).

	Median (Range) ^c^
Question on the satisfaction of …^a^	
the SelfMED procedure	6.0 (3–7)
the content of the questions in the nurses’ assessment	6.5 (6–7)
the content of the questions in the patient self-assessment	6.5 (6–8)
the number of questions in the nurses’ assessment	7.0 (7–8)
the number of questions in the patient self-assessment	7.0 (7–8)
the user-friendliness of the SelfMED procedure	6.0 (5–7)
the time investment to facilitate self-management of medication	4.0 (3–5)
The SelfMED procedure contributes to … ^b^	
better self-management of the patients’ prescribed medical therapy	6.5 (5–8)
better communication between patients, nurses, and physicians concerning the prescribed medical therapy	4.0 (3–5)
better understanding of patients’ competences to self-manage medication after hospital discharge	7.0 (6,7)

^a^ Ten-point scale; absolutely not satisfied—absolutely satisfied. ^b^ Ten-point scale; strongly disagree—strongly agree. ^c^ Full range.

## Data Availability

Data are available from the corresponding author upon reasonable request.

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
