# Peer review of "Self-Management of Medication on a Cardiology Ward: Feasibility and Safety of the SelfMED Intervention"

_ijerph, 2022, doi:10.3390/ijerph192416715_

Round 1
Reviewer 1 Report
1.- If the figures are your own, write it.
2.- "Nonparametric statistics were used"...List and justify which nonparametric statistics you used.
3.- Table 2. You must create two columns with the results of your statistical employees (Mann-Whitney U and Chi-Square test) as well as a third with the P values. It is the most important data of your study. It cannot appear as a footer. rebuild it.
4.- References 1 and 7 have neither DOI nor URL access.
5.- Check that all articles are freely accessible to readers of your article.
Reviewer 2 Report
In an era of increased demand for healthcare providers and the continuing crises of medicine misuse, the authors explore the possibility of patients’ self-managements of their medication in a controlled environment and possibly when they are discharged. The findings are interesting and serve as a pilot for large-scale clinical trials. Below are a few comments for the author's considerations
Line 16
Has the tool been used in a similar setting in the past?
Line 27
The authors might want to rethink the keywords used. below are some suggestions:
· Medicine safety
· Medication errors
· self-medication
Line 28
I think this article will benefit from expert proofreading. The language needs to be revised.
Line 45-50
Self-medication has associated risks that can lead to harm and even death in some cases. The author's account in the introduction section failed to elaborate on that which to me biased the discussion. I will recommend they review the literature and add to the section the implications of self-medications
Line 78
What is the rationale for choosing this set of patients and not others? what is the physical status of the patients in terms of vision, physical mobility, and alertness
Also, what type of medication was considered appropriate for this intervention and why? The authors might want to provide details of this information.
Line 112
How was the questionnaire designed and validated?
Line 162 – 163
Why some patients were only allowed t partly manage their medications?
Line 243 – 244
What is the implication of medication registration errors and how did the researchers strive to mitigate them in subsequent interventions?
Line 346
I think the authors should be cautious in making definitive conclusions because there are many limitations not accounted for. I suggest you revise the sentence and leave room for errors.
Reviewer 3 Report
Please see the attached file.
Thank you.
